# HyperTime: Implicit Neural Representations for Time Series

**Elizabeth Fons**
J. P. Morgan AI Research

**Alejandro Sztrajman**
University College London

**Yousef El-laham**
J. P. Morgan AI Research

**Alexandros Iosifidis**
Aarhus University

**Svitlana Vyetrenko**
J. P. Morgan AI Research

## Abstract

Implicit neural representations (INRs) have recently emerged as a powerful tool that provides an accurate and resolution-independent encoding of data. Their robustness as general approximators has been shown in a wide variety of data sources, with applications on image, sound, and 3D scene representation. However, little attention has been given to leveraging these architectures for the representation and analysis of time series data. In this paper, we analyze the representation of time series using INRs, comparing different activation functions in terms of reconstruction accuracy and training convergence speed. Secondly, we propose a hypernetwork architecture that leverages INRs to learn a compressed latent representation of an entire time series dataset. We introduce an FFT-based loss to guide training so that all frequencies are preserved in the time series. We show that this network can be used to encode time series as INRs, and their embeddings can be interpolated to generate new time series from existing ones. We evaluate our generative method by using it for data augmentation, and show that it is competitive against current state-of-the-art approaches for augmentation of time series.

## 1 Introduction

Modeling time series data has been a key topic of research for many years, constituting a crucial component of applications in a wide variety of areas such as climate modeling, medicine, biology, retail and finance [21]. Traditional methods for time series modeling have relied on parametric models informed by expert knowledge. However, the development of modern machine learning methods has provided purely data-driven techniques to learn temporal relationships. In particular, neural network-based methods have gained popularity in recent times, with applications on a wide range of tasks, such as time series classification [17], clustering [25, 2], segmentation [29, 43], anomaly detection [11, 40, 16], upsampling [28, 7], imputation [23, 24, 8], forecasting [21, 37] and synthesis [1, 41, 20]. In particular, the generation of time series data for augmentation has remained as an open problem, and is currently gaining interest due to the large number of potential applications such as in medical and financial datasets, where data cannot be shared, either for privacy reasons or for proprietary restrictions [19, 20, 4, 12].

In recent years, implicit neural representations (INRs) have gained popularity as an accurate and flexible method to parameterize signals, such as from image, video, audio and 3D scene data [32, 27]. Conventional methods for data encoding often rely on discrete representations, such as data grids, which are limited by their spatial resolution and present inherent discretization artifacts. In contrast, implicit neural representations encode data in terms of continuous functional relationships between signals, and thus are uncoupled to spatial resolution. In practical terms, INRs provide a new data representation framework that is resolution-independent, with many potential applications on time

NeurIPS 2022 Workshop on Synthetic Data for Empowering ML Research.

series data, where irregularly sampled and missing data are common occurrences [14]. However, there are currently no works exploring the suitability of INRs on time series representation and analysis.

In this work, we propose an implicit neural representation for univariate and multivariate time series data. We compare the performance of different activation functions in terms of reconstruction accuracy and training convergence. Finally, we combine these representations with a hypernetwork architecture, in order to learn a prior over the space of time series. The training of our hypernetwork takes into account the accurate reconstruction of both the time series signals and their respective power spectra. This motivates us to propose a Fourier-based loss that proves to be crucial in guiding the learning process. The advantage of employing such a Fourier-based loss is that it allows our hypernetwork to preserve all frequencies in the time series representation. In Section 4.2, we leverage the latent embeddings learned by the hypernetwork for the synthesis of new time series by interpolation, and show that our method performs competitively against recent state-of-the-art methods for time series augmentation.

## 2 Related Work

**Implicit Neural Representations**   Implicit Neural Representations (INRs) provide a continuous representation of multidimensional data, by encoding a functional relationship between input coordinates and signal values, avoiding possible discretization artifacts. They have recently gained popularity in visual computing [26, 27] due to the key development of positional encodings [36] and periodic activations (SIREN [32]), which have proven to be critical for the learning of high-frequency details. Whilst INRs have been shown to produce accurate reconstructions in a wide variety of data sources, such as video, images and audio [32, 10, 30], few works have leveraged them for time series representation [18, 39], and none have focused on generation.

**Hypernetworks**   Hypernetworks are neural network architectures that are trained to predict the parameters of secondary networks, referred to as Hyponetworks [15, 31]. In the last few years, some works have leveraged different hypernetwork architectures for the prediction of INR weights, in order to learn priors over image data [34] and 3D scene data [22, 33, 35]. [32] leverage a set encoder and a hypernetwork decoder to learn a prior over SIRENs encoding image data, and apply it for image in-painting. Our HyperTime architecture detailed in Section 3 uses a similar encoder-decoder structure, however we apply these architectures for time series generation via interpolation of learned embeddings.

**Time Series Generation**   Synthesis of time series data using deep generative models has been previously studied in the literature. Examples include the TimeGAN architecture [42], as well as QuantGAN [38]. More recently, [13] proposed TimeVAE as a variational autoencoder alternative to GAN-based time series generation. [1] introduced Fourier Flows, a normalizing flow model for time series data that leverages the frequency domain representation, which is currently considered together with TimeGAN as state-of-the-art for time series generation. In the last few years, multiple methods have used INRs for data generation, with applications on image synthesis [9, 34], super-resolution [10] and panorama synthesis [3]. However, there are currently no applications of INRs on the generation of time series data.

## 3 Formulation

In this Section we describe the network architectures that we use to encode time series data (Subsection 3.1), and the hypernetwork architecture (HyperTime) leveraged for prior learning and new data generation (Subsection 3.2).

### 3.1 Time Series Representation

In Figure 1 we present a diagram of the INR used for univariate time series. The network is composed of fully-connected layers of dimensions $1 \times 60 \times 60 \times 60 \times 1$, with sine activations (SIREN [32]):

$$\phi_i(\mathbf{x}_i) = \sin(\omega_0 \mathbf{W}_i \mathbf{x}_i + \mathbf{b}_i) \tag{1}$$

where $\phi_i$ corresponds to the $i^{th}$ layer of the network. A general factor $\omega_0$ multiplying the network weights determines the order of magnitude of the frequencies that will be used to encode the signal.

Input and output of the INR are uni-dimensional, and correspond to the time coordinate $t$ and the time series evaluation $f(t)$. Training of the network is done in a supervised manner, with MSE loss. After training, the network encodes a continuous representation of the functional relationship $f(t)$ for a single time series.

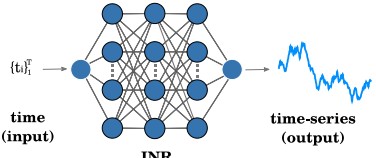

Figure 1: Diagram of the implicit neural representation (INR) for univariate time series. Neurons with a black border use sine activations.

The architecture from Figure 1 can be modified to encode multivariate time series, by simply increasing the number of neurons of the output layer to match the number of channels of the signal. Due to weight-sharing, this adds a potential for data compression of the time series.

## 3.2   Time Series Generation with HyperTime

In Figure 2 we display a diagram of the HyperTime architecture, which allows us to leverage INRs to learn priors over the space of time series. The Set Encoder (green network), composed of SIREN layers [32] with dimensions $2 \times 128 \times 128 \times 40$, takes as input a pair of values, corresponding to the time-coordinate $t$ and the time series signal $f(t)$. Each pair of input values is thus encoded into a full 40-values embedding and fed to the HyperNet decoder (blue network), composed of fully-connected layers with ReLU activations (MLP), with dimensions $40 \times 128 \times 7500$. The output of the HyperNet is a one-dimensional 7500-values embedding that contains the network weights of an INR which encodes the time series data from the input. The INR architecture used within HyperTime is the same described in the previous section, and illustrated in Figure 1. Following previous works [31], in order to avoid ambiguities we refer to these predicted INRs as HypoNets.

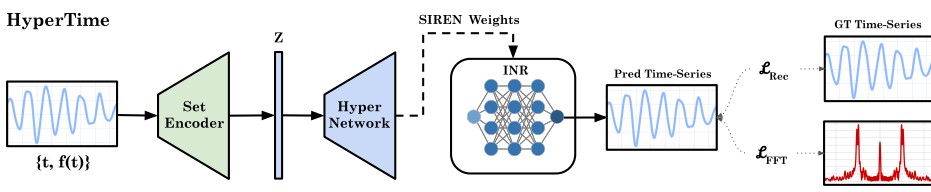

Figure 2: Diagram of HyperTime architecture. Each pair of time-coordinate $t$ and time series $f(t)$ is encoded by the Set Encoder. The HyperNet decoder learns to predict HypoNet weights from the embeddings. During training, the output of the HyperNet is used to build a HypoNet and evaluate it on in the input time-coordinates. The loss is computed as a difference between $f(t)$ and the output of the HypoNet $\hat{f}(t)$.

During the training of HyperTime, we use the weights predicted by the HyperNet decoder to instantiate a HypoNet and evaluate it on the input time-coordinate $t$, to produce the predicted time series value $\hat{f}(t)$. The entire chain of operations is implemented within the same differentiable pipeline, and hence the training loss can be computed as the difference between the ground truth time series signal $f(t)$ and the value predicted by the HypoNet $\hat{f}(t)$. After the training of HyperTime, the Set Encoder is able to generate latent embeddings $Z$ for entire time series. In Section 4.2, we show that these embeddings can be interpolated to synthesize new time series signals from known ones, which can be leveraged for data augmentation (see additional material for a pseudo-code of the procedure).

**Loss**   The training of HyperTime is done by optimizing the following loss, which contains an MSE reconstruction term $\mathcal{L}_{\text{rec}}$ and two regularization terms $\mathcal{L}_{\text{weights}}$ and $\mathcal{L}_{\text{latent}}$, for the network weights and the latent embeddings respectively:

$$\mathcal{L} = \underbrace{\frac{1}{N}\sum_{i=1}^{N}\left\|f(t_i) - \hat{f}(t_i)\right\|^2}_{\mathcal{L}_{\text{rec}}} + \lambda_1 \underbrace{\frac{1}{W}\sum_{j=1}^{W}w_j^2}_{\mathcal{L}_{\text{weights}}} + \lambda_2 \underbrace{\frac{1}{Z}\sum_{k=1}^{Z}z_k^2}_{\mathcal{L}_{\text{latent}}} + \lambda_3 \mathcal{L}_{\text{FFT}} \tag{2}$$

In addition, we introduce a Fourier-based loss $\mathcal{L}_{\text{FFT}}$ that focuses on the accurate reconstruction of the power spectrum of the ground truth signal (see Supplement for more details):

$$\mathcal{L}_{\text{FFT}} = \frac{1}{N} \sum_{i=1}^{N} \left\| \text{FFT}[f(t)]_i - \text{FFT}[\hat{f}(t)]_i \right\|. \tag{3}$$

In Section 4.2, we show that $\mathcal{L}_{\text{FFT}}$ is crucial for the accurate reconstruction of the time series signals.

## 4 Experiments

### 4.1 Reconstruction

Table 1: Comparison using MSE of implicit networks using different activation functions on different univariate and multivariate time series from the UCR dataset.

|  | Sine | ReLU | Tanh | Sigmoid |
|---|---|---|---|---|
| *Univariate* | | | | |
| Crop | **5.1e-06** | 5.4e-03 | 2.8e-02 | 5.1e-01 |
| NonInvasiveFetalECGThorax1 | **2.3e-05** | 2.8e-02 | 5.7e-02 | 8.1e-02 |
| PhalangesOutlinesCorrect | **7.5e-06** | 1.9e-02 | 1.4e-01 | 3.3e-01 |
| FordA | **9.2e-06** | 1.4e-01 | 1.5e-01 | 1.5e-01 |
| *Multivariate* | | | | |
| Cricket | **1.6e-04** | 4.2e-03 | 5.1e-03 | 1.6e-02 |
| DuckDuckGeese | **9.1e-05** | 8.0e-04 | 8.7e-04 | 9.1e-04 |
| MotorImagery | **1.7e-03** | 1.1e-02 | 1.1e-02 | 1.8e-02 |
| PhonemeSpectra | **1.1e-06** | 6.0e-03 | 1.6e-02 | 1.8e-02 |

We start by showing that encoding time series using SIRENS leads to a better reconstruction error than using implicit networks with other activations. We use univariate and multivariate time series datasets from the UCR archive [5].[1] We selected datasets with different characteristics, either short length time series or long, or in the case of the multivariate datasets, with many features (in some cases, more features than time series length). We sample 300 time series (or the maximum number available) from each dataset, train a single SIREN for each time series and calculate the reconstruction error. For comparison we train implicit networks using ReLU, Tanh and Sigmoid activations. As a sample case, we show in Figure 3 the losses and we observe that sine activations converge much faster, and to lower error values, than other activation functions.

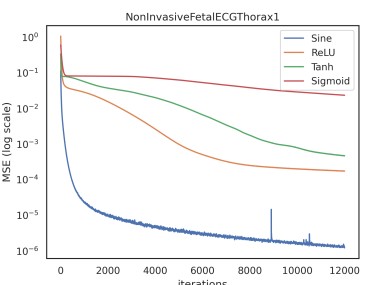

Figure 3: Comparison of MSE loss for implicit networks using different activation functions.

A summary of results can be found in Table 1, where we observe that the MSE error is at least an order of magnitude lower for sine activations, with respect to other activation layers.

### 4.2 Time Series Generation

To evaluate the utility of learning a prior over the space of implicit functions, we use the set encoder network and the hypernetwork to generate new time series. We do so by projecting time series into the latent vector of the HyperTime network and interpolating the latent vector. This is similar to training an autoencoder and interpolating the latent space, but the output of the decoder of HyperTime are the weights of the SIREN networks.

We follow the experimental set up proposed in [1] for the evaluation, were the performance of the synthetic data is evaluated using a predictive score (MAE) that corresponds to the prediction accuracy of an off-the-shelf neural network trained on the synthetic data and tested on the real data.

---

[1] The datasets can be downloaded from the project's website: `www.timeseriesclassification.com` [6]

Additionally, to measure the quality of the synthetic data, we use the precision and recall averaged over all time steps, which are then combined into a single F-score. We use the same datasets as before, and we add two datasets that were used in Fourier Flows [1] and TimeGAN [42], Google stocks data and UCI Energy data. We compare our HyperTime model with generating data using PCA, with Fourier Flows and TimeGAN, two state-of-the-art methods for time series generation. Table 2 shows the performance scores for all models and datasets. Additionally, we visualize

Table 2: Performance scores for data generated with Hyper-Time and for all baselines.

|  | Crop | NonInv | Phalanges | Energy | Stock |
|---|---|---|---|---|---|
| **PCA** | | | | | |
| *MAE* | 0.050 | 0.019 | 0.050 | **0.007** | 0.110 |
| *F1 Score* | **0.999** | **0.999** | **0.999** | 0.998 | **0.999** |
| **HyperTime (Ours)** | | | | | |
| *MAE* | **0.040** | **0.005** | **0.026** | 0.058 | 0.013 |
| *F1 Score* | **0.999** | 0.996 | 0.998 | **0.999** | 0.995 |
| **TimeGAN** | | | | | |
| *MAE* | 0.048 | 0.028 | 0.108 | 0.056 | 0.173 |
| *F1 Score* | 0.831 | 0.914 | 0.960 | 0.479 | 0.938 |
| **Fourier Flows** | | | | | |
| *MAE* | **0.040** | 0.018 | 0.056 | 0.029 | **0.008** |
| *F1 Score* | 0.991 | 0.990 | 0.992 | 0.945 | 0.992 |

the generated samples using t-SNE plots in Figure 4 where we can see that the generated data from HyperTime exhibits the same patterns as the original data. In the case of Fourier Flows, in the UCR datasets we see that NonInv and Phalanges do not show a good agreement.

The synthesis of time series via principal component analysis is performed in a similar fashion as our HyperTime generation pipeline. We apply PCA to generate a decomposition of time series into a basis of 40 principal components. The coefficients of these components constitute a latent representation for each time series of the dataset, and we can interpolate between embeddings of known time series to synthesize new ones. The main limitation of this procedure, besides its linearity, is that it can only be applied to datasets of equally sampled time series.

Finally, we analyze the importance of the Fourier-based loss $\mathcal{L}_{\text{FFT}}$ from equation 3 on the training of HyperTime. In Figure 5-left we display t-SNE visualizations of time series synthesized by HyperTime with and without the use of the FFT loss during training, for two datasets (NonInv and FordA). In both cases, the addition of the $\mathcal{L}_{\text{FFT}}$ loss results in an improved matching between ground truth and generated data. However, in the case of FordA, the addition of this loss becomes cru-

Table 3: Performance scores for data generated with HyperTime, with and without the Fourier-based loss $\mathcal{L}_{\text{FFT}}$, for two datasets (NonInv, FordA).

|  | NonInv | FordA |
|---|---|---|
| **HyperTime + FFT loss** | | |
| *MAE* | **0.0053** | **0.0076** |
| *F1 Score* | **0.9962** | **0.9987** |
| **HyperTime (no FFT)** | | |
| *MAE* | 0.0058 | 0.1647 |
| *F1 Score* | 0.9960 | 0.0167 |

cial to guide the learning process. This is also reflected in the numerical evaluations from Table 3, which shows steep improvements in performance for the FordA dataset.

A likely explanation for the difficulty of the network to learn meaningful patterns from the data of this dataset is provided by the right plot in Figure 5. Here we show the standard deviation of the power spectrum for both datasets, as a function of the frequency. The difference in the distributions indicates that FordA is composed of spectra that present larger variability, while NonInv's spectra are considerably more clustered. Further research on the characteristics of the datasets that benefit the more from the $\mathcal{L}_{\text{FFT}}$ loss should be further investigated, especially focusing on non-stationary time series.

## 5 Conclusions

In this paper we explored the use of implicit neural representations for the encoding and analysis of both univariate and multivariate time series data, and showed that periodic activation layers outperform traditional activations in terms of reconstruction accuracy and training speed. We presented HyperTime, a hypernetwork architecture to generate synthetic data which enforces not only learning an accurate reconstruction over the learned space of time series, but also preserving the shapes of the power distributions.

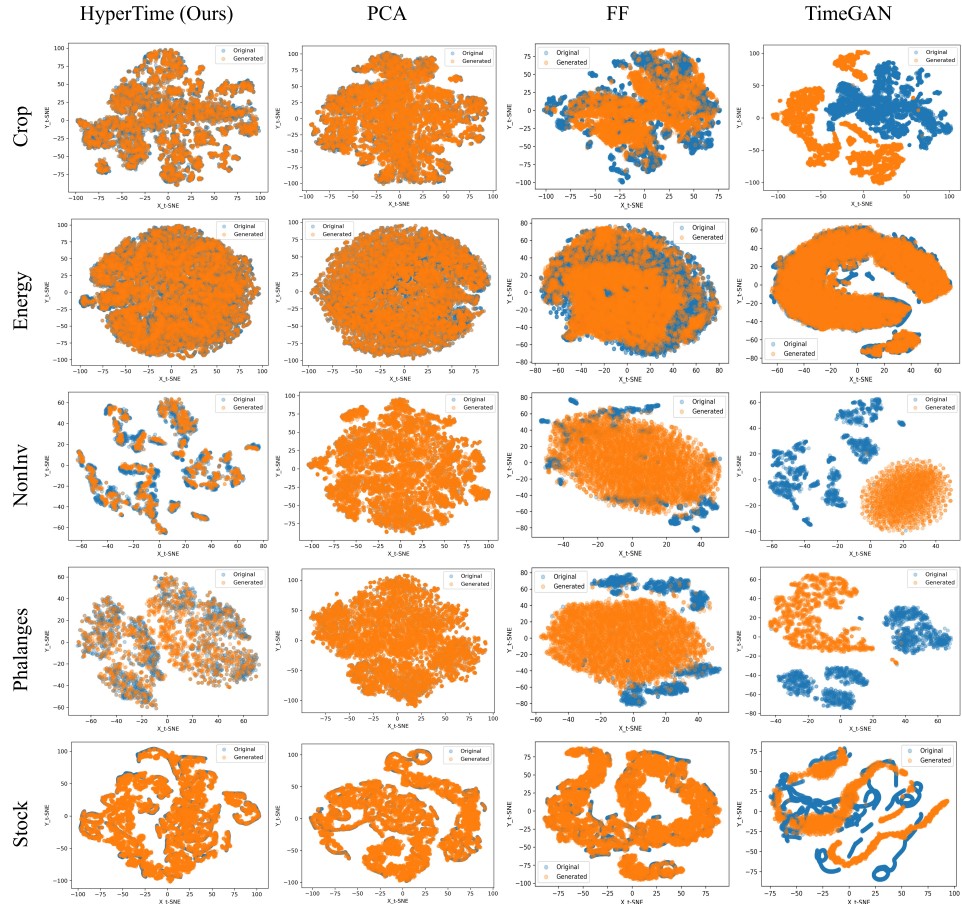

Figure 4: t-SNE visualization on univariate datasets (in rows: Stocks, Energy, Crop, NonInv and Phalanges), using different time series generation methods (in columns: HyperTime, PCA, Fourier Flows and TimeGAN). Blue corresponds to original data and orange to synthetic data.

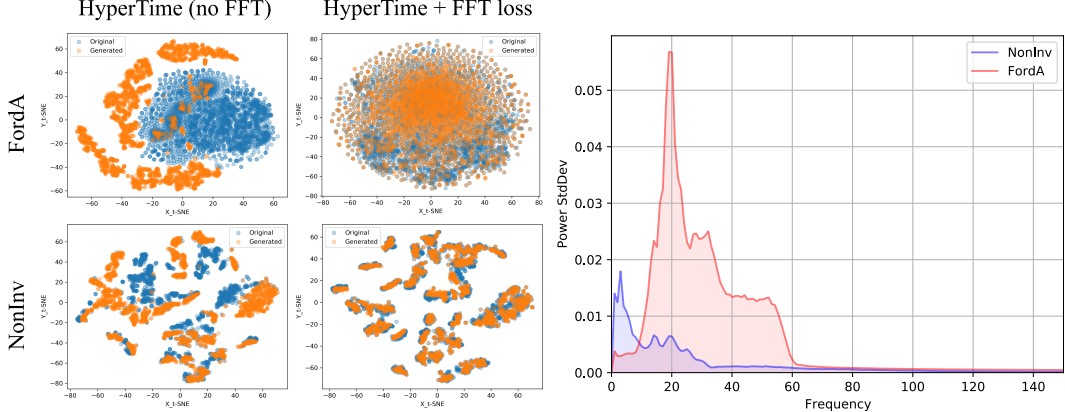

Figure 5: *Left:* t-SNE visualization of ground truth and generated data on two univariate datasets (NonInv and FordA), using HyperNet with and without the Fourier-based loss $\mathcal{L}_{\text{FFT}}$ (Eq. 3). *Right:* Standard deviation of the power spectra for the time series of the same two datasets. FordA shows a considerably larger number of variations in the distributions of the power spectra, which explains the difficulty of HyperTime to learn patterns from the data.

## Disclaimer

This paper was prepared for informational purposes in part by the Artificial Intelligence Research group of JPMorgan Chase & Coȧnd its affiliates ("JP Morgan"), and is not a product of the Research Department of JP Morgan. JP Morgan makes no representation and warranty whatsoever and disclaims all liability, for the completeness, accuracy or reliability of the information contained herein. This document is not intended as investment research or investment advice, or a recommendation, offer or solicitation for the purchase or sale of any security, financial instrument, financial product or service, or to be used in any way for evaluating the merits of participating in any transaction, and shall not constitute a solicitation under any jurisdiction or to any person, if such solicitation under such jurisdiction or to such person would be unlawful.

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
