# OpenReview forum: "HyperTime: Implicit Neural Representations for Time Series"
_NeurIPS.cc/2022/Workshop/SyntheticData4ML — Neurips 2022 SyntheticData4ML_

### Official Review · Reviewer_SDhy · 2022-10-12
**Review of "HyperTime: Implicit Neural Representations for Time Series" for NeurIPS 2022 Workshop SyntheticData4ML**

**Rating:** 6
**Confidence:** 4

**Review:**

1. Summary and contributions:

HyperTime uses implicit neural networks (INR) for generating time series. INR is treated as a Hyponetwork whose weights are estimated using a Hypernetwork. Authors assert that this is the first application of INR for generating time series. Besides, they incorporate a loss term for frequency content of time series using FFT. HyperTime is tested for a few datasets and its performance is compared with that of PCA, Fourier Flows, and TimeGAN. Results demonstrate an acceptable accuracy for the generation of time series.

2. Strengths:

Using INR for time series generation seems an appropriate choice as INR is intended for continuous data.

Incorporating frequency content of time series as a loss term is beneficial for the accurate reconstruction of time series.


3. Weaknesses:

Application of Hypernetwork for training INR was proposed in the following original work:
Sitzmann, V., Martel, J., Bergman, A., Lindell, D., & Wetzstein, G. (2020). Implicit neural representations with periodic activation functions. Advances in Neural Information Processing Systems, 33, 7462-7473.
Therefore, statements like “we propose a hypernetwork architecture that leverages INRs to learn a compressed latent representation of an entire time series dataset” purport a novelty that does not exist according to the prior work mentioned above.

Fourier transform represents global frequency content and dismisses local frequency characteristics of time series. Such weakness has led to the introduction of short-time Fourier Transform (STFT) and Wavelet Transform. If authors choose to use fourier transform over other transforms like STFT and Wavelet Transform, it should be justified or at least mentioned that future work may use alternate transforms.

Performance metrics are MAE and F-score. While MAE fails to punish large deviations of the generated data, the other metric, i.e. F1 score, averages precision and recall into a single score. Therefore, even if we assume that MAE correctly evaluates the precision, the diversity is not quantified separately. That being said, it is not clear that the F1-score is calculated based on which version of precision and recall. Note that the metrics in:
Sajjadi, M. S., Bachem, O., Lucic, M., Bousquet, O., & Gelly, S. (2018). Assessing generative models via precision and recall. Advances in neural information processing systems, 31.
are improved in:
Kynkäänniemi, T., Karras, T., Laine, S., Lehtinen, J., & Aila, T. (2019). Improved precision and recall metric for assessing generative models. Advances in Neural Information Processing Systems, 32.
The latter reference shows how the improved precision and recall can hardly be maximized simultaneously and one should seek a trade-off between them. Therefore, please provide the details for F1 score to be clear that it is based on which precision and recall metrics. It is suggested that the improved precision and recall are reported separately to distinguish between fidelity and diversity.

4. Correctness:
Not Applicable, there is no proof or code to evaluate correctness.

5. Clarity:
The language is clear.

6. Relation to prior work:
There is a dedicated section for related works which seems complete.

7. Reproducibility:
More details about datasets, training, hyperparameters, etc are needed for reproducibility.

8. Additional feedback, comments, suggestions for improvement and questions for the authors:
It is normal that reviewers search the literature to make sure that the work is novel to the extent it claims. Accordingly, searching “Implicit neural representations ”+”time series” in scholar gives an arxiv version of your work with the name of authors. The whole double-blind process is intended to reduce bias during the review, and to make reviewers rely on “what is said” rather than “who says what”. If authors do not collaborate to this end, double-blind review is not practical.

9.  Overall score:
(6/10)

10.  Confidence score:
(4/5)

11. Have the authors adequately addressed the broader impact of their work, including potential negative ethical and societal implications of their work?
No

12. Does the submission raise potential ethical concerns? This includes methods, applications, or data that create or reinforce unfair bias or that have a primary purpose of harm or injury. If so, please explain briefly.
One aspect of using synthetic data is preserving privacy. If the model generates data too similar to the original one, it may not be applicable in health and medical sectors, etc. Considering the overlap of generated data and original data, in Figure 4, the privacy aspect can be discussed.

14. Have you previously reviewed or area chaired (a version of) this work for another archival venue?
No

15. Agree to abide by the NeurIPS code of conduct?
Yes

---

### Official Review · Reviewer_7Daq · 2022-10-17
**Good work with a certain novelty, but the limitations left undiscussed cause concern.**

**Rating:** 6
**Confidence:** 3

**Review:**

The article is written soundly, well structured and easy to follow. Figure 2 practically summarizes the whole article: there is an autoencoder-like model, only for the decoder the authors use INR with weights generated from latent embeddings by a hypernetwork. Furthermore, they introduce an additional Fourier-based loss that focuses on accurate reconstruction, it is noticeably efficient, judging from the ablation study, is described in detail, and is probably the strongest part of the paper because of its clear novelty. The rest of the paper is devoted to combining known approaches for the new problem with results compatible with common solutions. In the experimental part, the authors tune INR with different activation functions, which can easily be moved to supplementary materials, and show that it is possible to generate synthetic data from latent embeddings good enough to be used as a training set for a separate classifier. The methodology of the experiments is solid, and I have no problem with this part or anything else written in the paper. The only immediate weakness is the hypernetwork, which have many limitations, starting from scalability and performance. The paper does not discuss the limitations, when they obviously should be at this point, and potential future work, nor the computational efficiency and speed of the proposed solution. If the proposed solution is much slower than the alternatives for the same performance, this is very important information.

---

### Official Review · Reviewer_ua2c · 2022-10-18
**Solid claim with sufficient results/evidence, great flow in writing with minor issues.**

**Rating:** 6
**Confidence:** 4

**Review:**

### Pros
- Paper discusses a novel attempt at using INR for time series data, FFT-based loss & INR embedding interpolation for time series generation.
- Paper is well written, with a thorough explanation of ideas, results, and evidence to support the claims.
- Ablation study was done on the FFT-based loss to show its impact.

### Cons
**Typos:**
- Line 165: "...using t-SNE plots 4..." --> "...using t-SNE plots in Figure 4..."

**Graphics:**
- Some fonts in tables and figures themselves seem a little small and hard to read.
- Black borders around the neurons in Figure 1 are REALLY hard to notice. I would highly recommend increasing the border thickness if possible.

### Other
1. Figure 3: The MSE loss from using Sine is visibly noisier than the other activation functions.
2. It is also a little interesting to see the Tanh and Sigmoid, the two activation functions commonly used in LSTMs are performing much worse.

Is there a possible explanation for these two scenarios mentioned above?

---

### Meta-Review · Area_Chair_N6hy · 2022-10-19

**Recommendation:** Accept